# Trends in fertility intentions and contraceptive practices in the context of COVID-19 in sub-Saharan Africa: insights from four national and regional population-based cohorts

Caroline Moreau [ID],[1,2] Celia Karp,[1] Shannon Wood [ID],[1] Kelsey Williams,[1] Funmilola M Olaolorun,[3] Pierre Akilimali,[4] Georges Guiella,[5] Peter Gichangi,[6] Linnea Zimmerman [ID],[1] Philip Anglewicz[1]

For numbered affiliations see end of article.

**Correspondence to**
Dr Caroline Moreau;
cmoreau2@jhu.edu

## ABSTRACT

**Objectives** Studies in several sub-Saharan geographies conducted early in the COVID-19 pandemic suggested little impact on contraceptive behaviours. Initial results may mask widening disparities with rising poverty, and changes to women's pregnancy desires and contraceptive use amid prolonged health service disruptions. This study examined trends in contraceptive behaviours in four sub-Saharan African settings 1 year into the pandemic.

**Design** Nationally and regionally representative longitudinal surveys.

**Setting** Burkina Faso, Kenya, Democratic Republic of Congo (Kinshasa) and Nigeria (Lagos).

**Participants** Women aged 15–49 years with sample size ranging from 1469 in Nigeria to 9477 in Kenya.

**Outcome measures** Fertility preferences, contraceptive use and unintended pregnancies measured before COVID-19 (November 2019 to January 2020) and during COVID-19 (November 2020 to January 2021).

**Analysis** We described population-level and individual-level changes by socioeconomic characteristics using generalised equation modelling. We used logistic regression models to identify factors related to contraceptive adoption and discontinuation and to experiencing an unintended pregnancy.

**Results** At the population level, we found no change in women's exposure to unintended pregnancy risk, alongside 5–9 percentage point increases in contraceptive prevalence in Burkina Faso, Kenya and Lagos. Reliance on provider-dependent methods dropped by 2 and 4 percentage points in Kenya and Burkina Faso, respectively, although these declines were not statistically significant. Between 1.0% and 2.8% of women across sites experienced an unintended pregnancy during COVID-19, with no significant change over time. Individual-level trajectories showed contraceptive adoption was more common than discontinuation in Burkina Faso, Kenya and Lagos, with little difference by sociodemographic characteristics. Women's COVID-19-related economic vulnerability was unrelated to unintended pregnancy across sites.

## STRENGTHS AND LIMITATIONS OF THIS STUDY

⇒ The study uses a longitudinal design to assess individual change in contraceptive behaviours since the onset of the COVID-19 pandemic.

⇒ Nationally or regionally representative survey data were used to evaluate population-level change in contraceptive behaviours since the onset of the COVID-19 pandemic.

⇒ This investigation spans four diverse geographies in sub-Saharan Africa, allowing for a comparison of COVID-19 impact by social and pandemic contexts.

⇒ Rates of loss to follow-up ranged from 21% to 27%, likely affecting study estimates, although the use of poststratification weights reduces attrition biases.

⇒ Data from two time points prevent the estimation of gaps in contraceptive use over the entire COVID-19 pandemic.

**Conclusions** This study highlights the resilience of African women across diverse settings in sustaining contraceptive practices amid the COVID-19 pandemic. However, with reports of rising poverty in sub-Saharan Africa, there is continued need to monitor access to essential sexual and reproductive health services.

## INTRODUCTION

The COVID-19 pandemic, which started in late 2019, quickly traversed the globe, creating health and economic crisis. By the end of 2020, the SARS-CoV-2 virus had resulted in nearly 2 million deaths,[1] destroyed 114 million jobs[2] and debilitated global economic growth rates, which fell to an annualised negative rate of −3.2.[2] With 65 468 confirmed fatalities, the sub-Saharan continent bore less of a health burden in this early stage of the pandemic,[1] but swift policy containment measures led to unprecedented economic loss, with a 1.9% contraction in the

BMJ

gross domestic product per capita and an estimated 32 million additional people living in extreme poverty.[3]

Loss of household income,[4] coupled with restricted access to essential health services, including family planning,[5 6] was expected to significantly alter fertility intentions and behaviours,[7–9] especially in sub-Saharan Africa,[10] given pre-existing high unmet need for sexual and reproductive health (SRH) services.[11] In 2019, the unintended pregnancy rate (91 per 1000 women per year) in sub-Saharan Africa was the highest in the world[12] and only 55% of women's need for contraception was satisfied.[13] However, initial empirical evidence from four sub-Saharan African countries, Burkina Faso, Democratic Republic of Congo (DRC), Kenya and Nigeria, using Performance Monitoring for Action (PMA) data, suggested little impact on women's use of contraception in the early stages of the pandemic.[14] In fact, the study found a rise in contraceptive prevalence in rural Burkina Faso and across Kenya. These data were corroborated with findings further exploring changes in women's individual-level contraceptive behaviours in the early months of the pandemic in Burkina Faso and Kenya, indicating that more women adopted contraception than discontinued their methods between the end of 2019 and mid-2020 (13% Burkina Faso and 25% Kenya vs 5% Burkina Faso and 6% Kenya, respectively).[15]

While these initial studies were pivotal in reporting early pandemic trends, the longer term societal impacts of COVID-19, including rising poverty[3 16] and interruptions to schooling and employment,[17] were likely to affect relationships and childbearing decisions.[18 19] At the same time, growing economic disparities were projected to affect people's financial access and use of health services,[4 20] potentially reducing their ability to achieve their reproductive goals. Furthermore, early impact estimates were drawn from phone-based surveys amid COVID-19 lockdowns, which may have affected the interpretation of early study findings due to selection bias and data collection mode.[14 15 21] Recent analysis estimating changes in modern contraception and pregnancy rates using repeated cross-sectional surveys collected face to face shed light on these trends. Specifically, in Burkina Faso, Kenya, Kinshasa (DRC) and Lagos (Nigeria), data collected in prepandemic periods and 1 year into the pandemic suggest slight decreases in pregnancy rates among younger, less educated women in Burkina Faso and a rise in modern contraception among young women in all geographies, except Lagos.[22]

Expanding on these results, the present study describes population-level and individual-level changes in women's fertility preferences, contraceptive behaviours and unintended pregnancy experiences in the first year of the COVID-19 pandemic using cross-sectional representative data and panel data collected in the same sub-Saharan African geographies. Specifically, we examined how these trends differed by sociodemographic characteristics and COVID-19-related economic hardship. Finally, we assessed factors related to individual-level changes in contraceptive behaviours and to experiencing an unintended pregnancy during the pandemic.

## METHODS

### Samples

We used two rounds of publicly available data (available at www.pmadata.org) collected in November 2019 to January 2020 (3–4 months before the pandemic) and 1 year later (November 2020 to January 2021) from nationally or regionally representative, population-based surveys. Surveys were implemented by the PMA project among women aged 15–49 years in four sub-Saharan African geographies. Surveys are nationally representative in Burkina Faso and Kenya, and regionally representative in Kinshasa province in DRC and Lagos state in Nigeria. We did not include data from Kongo Central in the DRC and Kano state in Nigeria, also collected through the PMA platform due to smaller sample size and lack of statistical power to show any reduction in contraceptive use in these two geographies.

Women were identified using a multistage clustered sampling design, starting with the random selection of census enumeration areas (EAs) after stratifying by rural and urban areas (in Kenya and Burkina Faso), followed by the random selection of 35 households per EA. All women of reproductive age (15–49 years) from the selected households were eligible for inclusion. Sample sizes were determined to produce modern contraceptive prevalence rates with a precision of ±3% at the national or regional level, leading to sample size variation by geography (table 1).

### Patient and public involvement

This study is a population-based study that does not include patients. The public was not involved in this research.

### Data collection

Prepandemic data (PMA's phase 1 surveys) were collected face to face by trained resident interviewers before COVID-19 (November 2019 to January 2020). Women who consented to be followed up as part of PMA's panel design were reinterviewed in person, 1 year later (PMA's phase 2 surveys), during the pandemic (November 2020 to January 2021). Replacement households from the EA were also included in phase 2 to account for attrition at the household level (projected at 15%). The survey design allows for longitudinal analysis to examine individual-level changes since the onset of COVID-19 and for annual cross-sectional analysis to track national or regional-level indicators (including household replacements). Follow-up rates between phase 1 and phase 2 surveys ranged from 73% in Kenya to 79% in Burkina Faso (table 1). No differences between phase 1 and phase 2 cross-sectional sample characteristics were observed; differential loss to follow-up in the panel sample was

**Table 1** Number of women aged 15–49 years included in each PMA survey and follow-up rate by setting

| Country/site | Phase 1 Baseline | Phase 2 Cross-sectional—including household replacement | Phase 2 Panel—no replacement sample (%) |
|---|---|---|---|
| Burkina Faso | 6590 | 6362 | 5207 (79) |
| Kenya | 9477 | 9310 | 6932 (73) |
| Kinshasa, DRC | 2611 | 2368 | 1967 (75) |
| Lagos, Nigeria | 1469 | 1481 | 1088 (74) |

Follow-up rate reported as % of all baseline women who successfully completed the phase 2 survey.
DRC, Democratic Republic of Congo; PMA, Performance Monitoring for Action.

accounted for by applying poststratification weights (more details available at www.pmadata.org).

### Measures

At each survey, women were asked about their current pregnancy status, whether the pregnancy was wanted or happened at the right time, future fertility intentions and current contraceptive behaviours. This information was operationalised as indicators used for population-level comparisons (cross-sectional measures) and for individual-level trends analysis (longitudinal measures).

### Cross-sectional measures

We explored six measures reflecting women's SRH experiences before and during the pandemic. *Women's fertility intentions for the next year* distinguished women who intended to have a child in the next year from those who wanted no more children or wanted to delay childbearing for more than 1 year. *Exposure to the risk of unintended pregnancy* (yes/no) classified women who had sexual intercourse in the last 12 months, were fecund and were not pregnant or trying to have a child in the next year as being 'at risk'. Among women at risk of unintended pregnancy, *use of contraception* (yes/no) classified women who were using barrier and natural methods, pills, injectables, patches, rings, implants, intrauterine devices (IUDs) and sterilisation as using contraception. Among contraceptive users, *use of a provider-dependent method* (yes/no) classified women using methods generally requiring interaction with a healthcare provider (ie, pills, injectables, implants, IUDs, sterilisation) as provider-dependent method users. We also considered a three-category measure of contraceptive method type, grouping contraceptive users according to the effectiveness of their contraceptive method—users of *less effective* methods (ie, barrier or natural methods), users of *effective* methods (ie, short-acting hormonal methods, including pills, injectables, patches or rings) and users of *highly effective* methods (ie, implants, IUDs or sterilisation). *Pregnancy status* identified women who were pregnant at the time of each survey (yes/no), while an indicator of *unintended pregnancy* (yes/no) identified women who were pregnant and indicated they had not wanted to become pregnant or wanted to become pregnant later.

### Longitudinal measures

We also explored individual-level changes in women's *exposure to the risk of unintended pregnancy, contraceptive use* and *experience of an unintended pregnancy* (reported as wanted later or not at all). Changes in women's exposure to unintended pregnancy risk across the two surveys were categorised into four categories: (1) *no risk*—never at risk before or during COVID-19; (2) *acquired risk*—newly at risk during COVID-19 (no risk before pandemic); (3) *ceased risk*—no longer at risk during COVID-19 (at risk before pandemic); (4) *sustained risk*—at risk before and during COVID-19. Likewise, we defined a four-category variable describing trends in any contraceptive use: (1) *continued non-use*—non-use before or during COVID-19; (2) *adoption*—during COVID-19 (non-use before pandemic); (3) *discontinuation*—non-use during COVID-19 following use before pandemic; (4) *continued use*—use before and during COVID-19. Finally, we considered the three-category indicator of contraceptive effectiveness (less effective/effective/highly effective) and a binary indicator of unintended pregnancy (yes/no) during phase 2 (ie, at the time of the COVID-19 survey), as previously defined.

We considered several sociodemographic and COVID-19-related factors that could impact women's use of contraception. Specifically, we assessed age, parity, marital status (in union/not in union), education, household wealth in tertiles and residence (urban/rural). During COVID-19, we also considered two indicators of household economic vulnerability: loss of income in the last 12 months (categorical—none, partial, complete) and food insecurity (ie, not being able to eat for 24 hours due to lack of food since the beginning of the COVID-19 pandemic).

### Analysis

We first examined population-level trends in reproductive health indicators comparing phase 1 and phase 2 cross-sectional samples across our six indicators (proportion who intend to have a child in the next year, proportion at risk of unintended pregnancy, proportion using contraception among women who were at risk of unintended pregnancy, proportion using provider-dependent methods among users, proportion pregnant and proportion experiencing an unintended pregnancy).

We conducted the same analysis stratified by rural and urban residence in Burkina Faso and Kenya to compare trends across urban localities (urban Burkina Faso, urban Kenya, Lagos-Nigeria and Kinshasa-DRC and rural localities in Burkina Faso and Kenya). Statistically significant differences were determined based on non-overlapping CIs between phase 1 and phase 2 estimates.

Next, we conducted individual-level analysis, focusing on the sample of women completing both surveys. We evaluated changes in exposure to unintended pregnancy risk and changes in contraceptive behaviours (among women at risk at both time points). We used generalised estimation equation (GEE) models to test for differential trends in the use of any contraception between phase 1 and phase 2 by women's sociodemographic characteristics (measured at phase 1) and their experience of economic instability (measured at phase 2) (online supplemental appendix 1). Next, we examined the contraceptive method mix of contraceptive adopters and the methods women discontinued between the two surveys. We conducted three multivariable logistic regression models to identify factors related to (1) contraceptive adoption among women who were not using contraception before pandemic, (2) contraceptive discontinuation among women who were using a method before pandemic and (3) the experience of an unintended pregnancy during the pandemic.

All analyses applied survey and poststratification weights to account for the complex survey design, clustering of women within EAs and differential loss to follow-up in the case of the longitudinal sample. All analyses were conducted in Stata V.16.1 with statistical significance set a priori at $p<0.05$.

## RESULTS

The characteristics of prepandemic study samples by setting are presented in table 2. By design, all women in Kinshasa-DRC and Lagos-Nigeria were living in urban areas, while this was the case for 22.8% and 30.2% of the samples in Burkina Faso and Kenya, respectively. The mean age of women ranged from 28.3 years in Kinshasa-DRC to 31.0

| Table 2 | Phase 1 sociodemographic characteristics by setting | | | | |
|---|---|---|---|---|---|
| | | Burkina Faso (n=6590) | Kenya (n=9477) | Kinshasa-DRC (n=2611) | Lagos-Nigeria (n=1469) |
| Age (mean) | | 28.9 | 28.8 | 28.3 | 31.0 |
| | | Column % | | | |
| Age | 15–19 | 21.6 | 21.6 | 22.2 | 14.0 |
| | 20–24 | 17.3 | 17.6 | 20.9 | 14.0 |
| | 25–29 | 14.7 | 16.3 | 16.3 | 15.5 |
| | 30–34 | 15.6 | 15.4 | 12.3 | 17.7 |
| | 35–39 | 12.7 | 11.8 | 10.7 | 17.8 |
| | 40–44 | 10.9 | 9.5 | 10.4 | 12.7 |
| | 45–49 | 7.2 | 7.8 | 7.2 | 8.3 |
| Union status | In union* | 75.8 | 59.2 | 42.7 | 60.9 |
| | Not in union | 24.2 | 40.8 | 57.3 | 39.1 |
| Parity | Nulliparous | 24.3 | 28.3 | 41.9 | 35.2 |
| | 1–2 children | 24.8 | 30.9 | 28.1 | 29.5 |
| | 3+ children | 50.9 | 40.9 | 30.1 | 35.3 |
| Residence | Urban | 22.8 | 30.2 | 100.0 | 100.0 |
| | Rural | 77.2 | 69.8 | – | – |
| Education | No schooling/primary | 77.6 | 49.7 | 7.8 | 11.8 |
| | Secondary low | 16.4 | 2.1 | 72.9 | 51.2 |
| | Secondary high or more | 6.1 | 48.2 | 19.3 | 37.0 |
| Contraception† | None | 70.1 | 52.2 | 55.4 | 60.6 |
| | Less effective | 5.7 | 7.2 | 32.7 | 25.8 |
| | Effective | 9.9 | 19.2 | 4.2 | 6.1 |
| | Highly effective | 14.3 | 21.4 | 7.8 | 7.5 |

*In union defined as married or living with a partner as if married.
†Less effective=barrier or natural methods; effective=short-acting hormonal methods, including pills, injectables, patches or rings; highly effective=IUDs, implants, sterilisation.
DRC, Democratic Republic of Congo; IUD, intrauterine device.

**Table 3** Cross-sectional reproductive characteristics of women at phase 1 and phase 2 by setting

| | Burkina Faso | | Kenya | | DRC-Kinshasa | | Nigeria-Lagos | |
|---|---|---|---|---|---|---|---|---|
| | Phase 1 (n=6590) | Phase 2 (n=6362) | Phase 1 (n=9477) | Phase 2 (n=9310) | Phase 1 (n=2611) | Phase 2 (n=2368) | Phase 1 (n=1469) | Phase 2 (n=1481) |
| | % (95% CI) | | | | | | | |
| Intends to have a child in the next year | 13.4 (11.8 to 15.3) | 14.1 (12.3 to 16.1) | 6.9 (6.1 to 7.8) | 6.2 (5.5 to 6.9) | 12.2 (9.3 to 15.8) | 10.1 (7.8 to 13.2) | 16.8 (14.6 to 19.2) | 16.2 (14.6 to 18.1) |
| In need of contraception | 44.1 (41.1 to 47.2) | 44.5 (41.4 to 47.4) | 46.3 (44.9 to 47.6) | 45.6 (44.3 to 46.9) | 27.8 (24.7 to 31.1) | 29.2 (26.3 to 32.2) | 39.5 (36.6 to 42.5) | 40.4 (37.7 to 43.2) |
| Contraceptive use among women in need | **43.2 (39.0 to 47.6)** | **52.3 (48.0 to 56.6)** | **71.3 (69.4 to 73.2)** | **76.5 (74.6 to 78.2)** | 71.8 (67.4 to 75.9) | 72.4 (68.2 to 76.2) | 63.4 (56.8 to 69.5) | 69.7 (63.7 to 75.2) |
| Provider-dependent contraception among contraceptive users | 83.4 (79.3 to 86.9) | 79.1 (74.3 to 83.1) | 88.3 (86.7 to 89.8) | 86.3 (84.6 to 87.8) | 37.3 (29.5 to 45.9) | 35.6 (28.3 to 43.5) | 40.1 (33.7 to 46.9) | 35.2 (29.1 to 41.8) |
| Currently pregnant | 8.5 (7.6 to 9.6) | 7.6 (6.6 to 8.4) | 5.4 (4.9 to 5.9) | 5.1 (4.6 to 5.7) | 5.7 (4.5 to 7.1) | 5.2 (4.1 to 6.5) | 4.7 (3.7 to 6.0) | 4.7 (3.7 to 5.9) |
| Current unintended pregnancy | 2.8 (2.1 to3.7) | 2.1 (1.6 to2.7) | 2.3 (2.0 to2.7) | 2.1 (1.7 to2.4) | 2.6 (1.9 to3.6) | 2.8 (2.1 to3.8) | 1.0 (0.6 to 1.6) | 1.0 (0.6.1.7) |

Numbers in bold indicate statistical significance based on the non-overlap of CIs.
DRC, Democratic Republic of Congo.

years in Lagos-Nigeria. Most women across settings were in union, except in Kinshasa, where this was the case in 42.7% of respondents. Women in Burkina Faso had the highest parity (50.9% with three or more children), followed by Kenya (40.9% with three or more children), while lower parities were observed in Kinshasa and Lagos, where 41.9% and 35.2% of women, respectively, had no children. Educational attainment varied substantially across settings, as 77.6% of women in Burkina Faso had no schooling or a primary level of education, with this proportion dropping to 49.7% in Kenya and to 7.8% and 11.8% in Kinshasa-DRC and Lagos-Nigeria, respectively. Contraceptive practices also varied, with non-use ranging from 52.2% in Kenya to 70.1% in Burkina Faso. Reliance on highly effective methods was greatest in Kenya (21.4%) and Burkina Faso (14.3%), while use of less effective barrier and natural methods was more common in Kinshasa-DRC (32.7%) and Lagos-Nigeria (25.8%).

At the population level, 6.2%–16.8% of women intended a birth in the next 12 months across geographies, with no significant change during the pandemic (table 3), including in urban and rural areas (online supplemental appendix 2). Accounting for past-year sexual activity, fertility intentions and fecundity, 27.8%–46.3% of women were exposed to the risk of unintended pregnancy, with the highest proportion observed in Burkina Faso. Exposure to unintended pregnancy risk did not vary over time in any setting.

Contraceptive use (any method) varied widely by setting and was lowest in Burkina Faso and highest in Kenya and Kinshasa province in the DRC. During the pandemic,

contraceptive use rose significantly by 9 percentage points in Burkina Faso and 5 percentage points in Kenya. The rise in contraceptive use among women in Burkina Faso was twice as elevated in rural settings (10 percentage points) compared with urban settings (5 percentage points; non-significant difference) while increases were comparable and statistically significant in rural and urban Kenya (online supplemental appendix 2). Contraceptive use also rose by 6 percentage points in Lagos-Nigeria (non-significant difference), while it remained stable in Kinshasa-DRC. The proportion of contraceptive users relying on provider-dependent methods ranged from 35.2% in Lagos to 88.3% in Kenya. This proportion remained stable in Kinshasa-DRC and Lagos-Nigeria and declines were non-significant in Burkina Faso and in Kenya overall (overlapping CIs) (table 3), but dropped by a significant 3 percentage points in rural Kenya (from 92.3% to 89.1%, online supplemental appendix 2).

Between 5.1% of women in Kenya and 8.5% of women in Burkina Faso were pregnant at the time of the survey with no change over time. Finally, between 1.0% and 2.8% of women across sites had an unintended pregnancy at the time of the phase 2 survey, this proportion remaining stable over time in all sites (table 3). No significant differences in trends were observed between women in urban and rural settings (online supplemental appendix 2).

Assessment of individual-level trajectories of women who responded to both surveys indicated that changes in exposure to unintended pregnancy risk among all women ranged from 16.9% in Kinshasa-DRC to 31.5% in Burkina Faso (figure 1). Patterns of change were similar

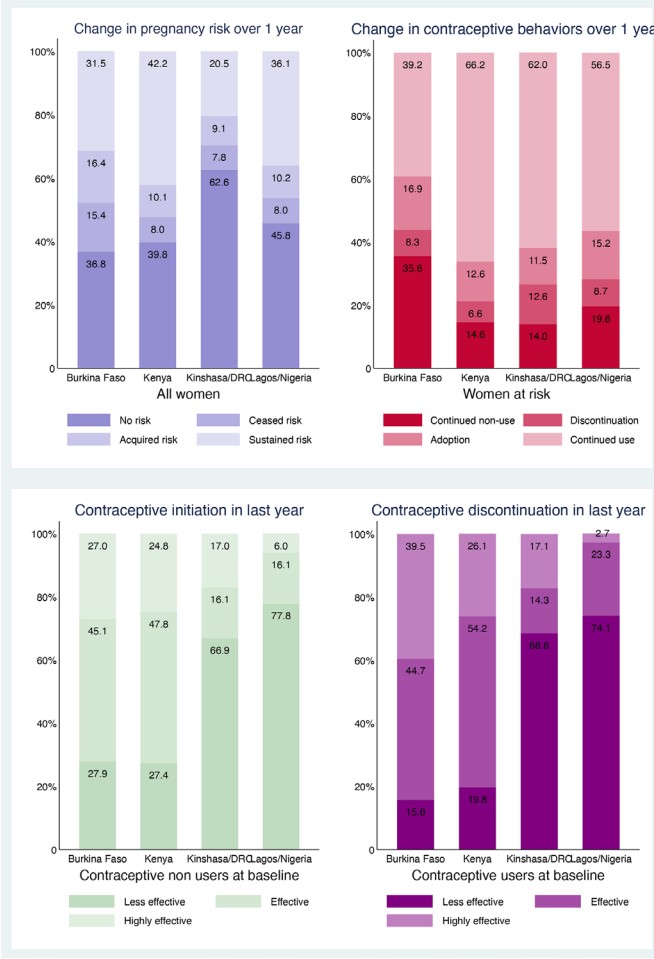

**Figure 1** Changes in individual-level risk of unintended pregnancy, and use of contraception between phase 1 and phase 2, and contraceptive method mix between adopters and discontinuers of contraception. DRC, Democratic Republic of Congo.

in rural versus urban settings in Burkina Faso and Kenya (data not shown). Half of changes involved new exposure to risk, while the other half led to cessation of exposure to risk during the pandemic. Among all women, the proportion at risk of unintended pregnancy at both time points (ie, sustained risk) ranged from 20.5% in Kinshasa-DRC to 42.2% in Kenya. Among women with sustained risk, change in contraceptive status varied from 19.4% in Kenya to 25.2% in Burkina Faso. Contraceptive adoption was more common than discontinuation in Burkina Faso (16.9% vs 8.3%, respectively), Kenya (12.6% vs 6.8%, respectively) and Lagos-Nigeria (15.2% vs 8.7%, respectively) but equivalent in Kinshasa-DRC (11.5% vs 12.6%, respectively). Among women who changed contraceptive status, most adopted or discontinued less effective barrier and natural methods in Kinshasa-DRC (66.9% and 68.5%, respectively) and Lagos-Nigeria (77.8% and 74.1%, respectively), while most adopted or discontinued effective short-acting methods in Burkina Faso (45.1% and 44.7%, respectively) and Kenya (47.8% and 54.2%, respectively). Highly effective long-acting method

contributed the least to adoption and discontinuation in Lagos-Nigeria (6.0% and 2.7%, respectively) and accounted for 27.0% of adoption and 39.5% of discontinuation in Burkina Faso. Patterns of change reflect differences in contraceptive method mix among women at risk before COVID-19, as 65.5% and 73.2% of contraceptive users relied on less effective methods in Kinshasa-DRC and Lagos-Nigeria, respectively, versus 15% in Kenya and 19% in Burkina Faso. Conversely, 44.7% of women at risk in Kenya and 47.8% in Burkina Faso relied on highly effective methods before the pandemic versus 17.4% in Kinshasa-DRC and 19.0% in Lagos-Nigeria.

While results from GEE models identified few statistically significant interactions between time and women's characteristics, signalling limited differences in contraceptive trends by sociodemographic characteristics, some patterns emerged (table 4). For example, increases in the odds of contraceptive use during COVID-19 relative to the prepandemic period were more pronounced among women in their mid-20s compared with teenagers or women over 40 years in all sites but Kinshasa-DRC. Likewise, increases in the odds of contraceptive use were elevated among parous women but not among nulliparous women in Kenya. In Lagos-Nigeria, increases in the odds of contraceptive use were greater among women with one to two children relative to those who had more children. Finally, wealthiest women in Lagos experienced greater increases in the odds of contraceptive use relative to women in the medium wealth category in Lagos. Conversely, there were no differential trends by education, recent loss of income or recent experience of food insecurity.

Few factors were related to contraceptive adoption among women who were not using a method before the pandemic and to contraceptive discontinuation among those using contraception before the pandemic (online supplemental appendix 3). The odds of adopting contraception were higher among women with secondary education or higher in Burkina Faso and Kenya, relative to less educated women, and among women in the richest tertile, relative to the poorest women in Kenya. Conversely, the oldest women (35–49 years) were less likely to adopt a method in Kinshasa-DRC and Kenya and more likely to discontinue their method in Kenya. Discontinuation was also more common among parous women compared with nulliparous women in Kenya. Finally, women with lower secondary education in Kinshasa were less likely to discontinue their method than women with no schooling or primary education.

Finally, factors associated with characterising an unintended pregnancy during COVID-19 among women who had sexual intercourse in the last 12 months are reported in online supplemental appendix 4. Older women (35 years or older) and women using long-acting contraception were less likely to experience an unintended pregnancy across sites. Women with higher education and from wealthier households were less likely to experience an unintended pregnancy in Kenya, while women of

**Table 4** Differential trends in contraceptive use by women's sociodemographic characteristics

| | Burkina Faso (n=5207) | Kenya (n=6932) | DRC-Kinshasa (n=1967) | Nigeria-Lagos (n=1088) |
|---|---|---|---|---|
| | RRR (95% CI) | | | |
| Overall | 1.4 (1.2 to 1.6) | 1.3 (1.2 to 1.5) | 0.9 (0.7 to 1.2) | 1.3 (1.1 to 1.6) |
| Age#by year interaction* | P=0.06 | P=0.24 | P=0.13 | P=0.08 |
| 15–19 | 1.3 (0.8 to 2.2) | 1.4 (0.7 to 2.8) | 0.4 (0.1 to2.6) | – |
| 20–24 | 1.4 (1.1 to 1.8) | 1.3 (1.0 to 1.7) | 1.5 (0.6 to3.6) | 1.1 (0.5 to 2.2) |
| 25–29 | 1.7 (1.2 to 2.3) | 1.5 (1.2 to 1.9) | 1.2 (0.7 to2.2) | 2.9 (1.5 to 5.6) |
| 30–34 | 1.2 (0.8 to 1.6) | 1.4 (1.2 to 1.6) | 0.4 (0.2 to0.7) | 1.7 (1.0 to 3.1) |
| 35–39 | 1.6 (1.2 to 2.3) | 1.5 (1.2 to 1.8) | 1.1 (0.6 to1.8) | 1.0 (0.7 to 1.4) |
| 40–44 | 1.5 (1.0 to 2.1 | 1.1 (0.9 to 1.3) | 0.9 (0.6 to1.4) | 1.2 (0.7 to 2.1) |
| 45–49 | 0.8 (0.7 to 1.1) | 1.1 (0.9 to 1.4) | 1.0 (0.6 to1.7) | 1.1 (0.6 to 2.0) |
| Parity#year interaction* | P=0.34 | P=0.08 | P=0.88 | P=0.004 |
| Nulliparous | – | 0.7 (0.2 to 2.1) | 3.3 (0.2 to 55.9) | – |
| 1–2 | 1.4 (1.1 to 1.8) | 1.5 (1.2 to 1.8) | 0.8 (0.5 to 1.3) | 2.1 (1.4 to 23.0) |
| 3+ | 1.4 (1.2 to 1.6) | 1.3 (1.1 to 1.4) | 1.0 (0.7 to 1.2) | 1.1 (0.8 to 1.4) |
| Residence#year interaction* | P=0.32 | P=0.54 | | |
| Urban | 1.3 (1.1 to 1.4) | 1.2 (1.0 to 1.5) | – | – |
| Rural | 1.4 (1.2 to 1.6) | 1.3 (1.2 to 1.5) | – | – |
| Wealth#year interaction* | P=0.57 | P=0.61 | P=0.42 | P=0.08 |
| Poorest | 1.5 (1.2 to 2.0) | 1.4 (1.2 to 1.6) | 0.9 (0.5 to 1.4) | 1.5 (0.9 to 2.3) |
| Medium | 1.3 (1.0 to 1.6) | 1.3 (1.1 to 1.6) | 1.1 (0.7 to 1.8) | 1.1 (0.7 to 1.6) |
| Richest | 1.3 (0.9 to 1.3) | 1.2 (0.9 to 1.5) | 1.1 (0.7 to 1.8) | 2.0 (1.3 to 2.9) |
| Loss of income last 12 months | P=0.30 | P=0.48 | P=0.71 | P=0.09 |
| None | – | 1.2 (1.0 to 1.4) | – | – |
| Partial | 1.4 (1.2 to 1.6) | 1.4 (1.2 to 1.5) | 0.9 (0.7 to1.3) | 1.4 (1.1 to 1.9) |
| Complete | 1.5 (1.0 to 2.3) | 1.3 (1.1 to 1.5) | 0.9 (0.7 to 1.2) | 1.2 (0.8 to 1.7) |
| Food insecurity | P=0.24 | P=0.33 | P=0.85 | P=0.10 |
| No | 1.4 (1.2 to 1.5) | 1.3 (1.2 to 1.4) | 1.0 (0.7 to 1.3) | 1.4 (1.1 to 1.7) |
| Yes | 1.8 (1.1 to 2.9) | 1.5 (1.2 to 1.9) | 0.7 (0.5 to 1.1) | 0.8 (0.6 to 1.1) |
| Education | P=0.29 | P=0.87 | P=0.26 | P=0.22 |
| No schooling/primary | 1.4 (1.2 to 1.6) | 1.3 (1.2 to 1.4) | 0.6 (0.3 to 1.3) | 1.2 (0.7 to 2.0) |
| Secondary low | 1.3 (0.8 to 2.0) | 1.2 (0.6 to 2.4) | 1.0 (0.8 to 1.3) | 1.2 (0.9 to 1.7) |
| Secondary high or higher | 0.7 (0.3 to 1.3) | 1.3 (1.1 to 1.6) | 0.7 (0.4 to 1.2) | 1.9 (1.4 to 2.6) |

*P values refer to the significance of the interaction term between all categories of a demographic factor and the outcome of interest (contraceptive use).
DRC, Democratic Republic of Congo; RRR, Relative risk Ratio.

higher parity were more likely to experience unintended pregnancies in Burkina Faso and Kinshasa-DRC. COVID-19-related economic vulnerability was unrelated to unintended pregnancy across sites.

## DISCUSSION

This study confirms women's resiliency in contraceptive use to prevent unintended pregnancies across a diversity of sub-Saharan contexts, despite the profound economic and social disruptions caused by the COVID-19 pandemic.[2–4 16 17] While we found a sustained need for contraception, we also observed an increase in contraceptive coverage in three of the four geographies studied. Within Burkina Faso and Kenya, increases in contraceptive protection corresponded with a decrease in the use of methods requiring a prescription, which was only significant in rural Kenya, but most women still used effective or highly effective methods, including women who adopted methods during the pandemic. These contraceptive dynamics coincide with the absence of significant

changes in pregnancy rates or unintended pregnancy rates across sites.

These results challenge the initial forecasts of rising unintended pregnancies due to service disruptions to SRH services in low and middle-income countries.[7] The absence of negative population-level effects of the pandemic on contraceptive behaviours and reproductive outcomes, noted in a previous study focusing on the early months of the pandemic,[14 15] and confirmed in a study using 1-year follow-up data,[22] is surprisingly reassuring and suggests several potential explanations.[23]

First, while disruptions to SRH services were reported in the early stage of the pandemic,[5 23–28] patterns of service disruptions varied significantly by level, duration, type of service and geography.[24–26] Discontinuity of services was often short lived—several studies reporting rapid post-lockdown rebounds in service provision.[23 27–29]

Further, it is possible that the quality of facility-based registers, including clients' family planning records, was impacted due to understaffing amid COVID-19-related disruptions and, thus, early pandemic provision of care was under-reported.[30]

At the same time, the expansion or continued use of high-impact practices in family planning,[31] such as community health workers, mobile outreach, supply chain management and digital health initiatives, may have compensated for facility-based disruptions to care. For example, the expansion of community health workers to deliver family planning services resulted in a 2.5% increase in the number of family planning clients in rural Kenya in the summer of 2020.[32] These initiatives were encouraged by the Kenyan government who issued guidelines to support 'Continuity of Reproductive, Maternal, Newborn and Family Planning Care and Services in the Background of COVID-19 Pandemic' in the month following the first reported case of COVID-19.[33] More generally, a recent WHO report of SRH services in Africa during COVID-19 indicates that 15 out of the 17 countries reflected in the survey had integrated SRH services in their continuity of care plan during COVID-19, including awareness campaigns and communication messages about family planning.[25]

A third explanation lies in the resilience of women in intensifying their efforts or finding alternative solutions to access contraception and self-manage their contraceptive needs.[24] This hypothesis is plausible given that our initial investigation into family planning amid COVID-19, conducted between May and July 2020, found that only 3.8%–14.4% of women at risk of unintended pregnancy were not using contraception for COVID-19-related reasons in Burkina Faso and Kenya; these women primarily cited fear of COVID-19 transmission.[15] Approximately 6 months later, the data collected via PMA surveys between November 2020 and January 2021 included in the present analysis reveal 9%–35% reported difficulties accessing any health service in the last year; however, 89%–93% of them were ultimately successful in obtaining services (www.pmadata.org). Self-provision

of contraceptive pill/oral contraceptives during the COVID-19 is likely to have facilitated women's access to short-acting contraception,[23] while the impact of policies allowing self-administration of injectable contraception should be further investigated to evaluate their contribution to continuity of family planning use during the pandemic.

These overall trends remain encouraging, but a critical concern over widening social disparities in family planning access may still arise in the face of increasing economic hardships.[3 4] Our findings provide no evidence of adverse SRH outcomes among women who report food insecurity, or those who experienced substantial loss of income since the start of the COVID-19 pandemic. Recent FP2030 policy commitments towards family planning (https://fp2030.org/FP2030-commitment-makers), including increased public funding, and expansion of free access to contraception in some countries, such as Burkina Faso in 2019, are likely to have mitigated the economic impact of COVID-19, but the affordability of contraception remains a prominent concern for women experiencing economic hardship.[18]

Our results should be considered with several limitations in mind. Specifically, the loss to follow-up rate ranged from 21% to 27% across geographies, which is likely to affect the reported estimates of change. However, the use of replacement households in phase 2 of data collection was designed to minimise impacts to population-level estimates. Further, poststratification weights were used in panel analysis to limit the potential for attrition biases; stable estimates of sociodemographic distributions across samples were observed (online supplemental appendix 5). A second limitation is our measure of change using two time points, which prevents a more comprehensive assessment of changes over the full-time period, including potential gaps in contraceptive use. A further evaluation of contraceptive use dynamics, using calendar data, was beyond the scope of this analysis but could provide a more thorough investigation of contraceptive gaps, although limited by the quality of the calendar data.[34] Our data in DRC and Nigeria are not nationally representative but rather reflect urban trends. Stratified analyses comparing urban and rural Burkina Faso and Kenya found no major differences in family planning trends; however, these results cannot be generalised to DRC and Nigeria, given distinct patterns of SRH disruptions reported in several studies in Africa.[25 26] Finally, our sample sizes were limited to detect significant trends in rare events, such as pregnancy outcomes, or differential trends by women's sociodemographic and economic background. In addition, a more thorough investigation of changing fertility intentions and their impact on subsequent reproductive outcomes over the course of the pandemic could shed more light on the ways fertility responds to external shocks.

Our study also has several strengths. First, our population-based approach provides a complimentary perspective to data collected via Health Management Information

Systems, showing little change in behaviours and SRH outcomes, despite facility-based disruptions, suggesting the importance of non-facility-based services and self-care practices in ensuring continuity of family planning care.[25] Second, our novel study design, integrating perspectives from representative cross-sectional data and panel data, offers the double advantage of examining population and individual trends between pre–COVID-19 and COVID-19 periods, with the potential of identifying disparities in COVID-19 effects on family planning practices. The use of a comprehensive set of indicators ranging from pregnancy exposure to contraceptive practices and reproductive outcomes also provides a more holistic understanding of fertility processes during a pandemic, elucidating how family planning needs and use evolve concurrently. In particular, our focus on unintended pregnancy provides added information about women's ability to achieve their reproductive goals in the context of the global pandemic. Finally, the cross-national/regional nature of the study, enhanced by using a common study protocol and indicators across geographies, strengthens comparative trends analysis, supporting previously reported variations in COVID-19 effects on SRH services.[25 26]

In conclusion, the current study provides important insights on the resilience of women across geographically and socially diverse contexts in sub-Saharan Africa in sustaining contraceptive practices amid the COVID-19 pandemic. Renewed national and international commitments to family planning made as part of the 2011 Ouagadougou Partnership[35] and 2012 London Summit on Family Planning[36] have likely contributed to the resiliency and innovations in family planning delivery systems. The integration of maternal health and other reproductive health services into continuity of care plans during COVID-19 is a reflection of such commitments.[25] However, investments in these systems must be sustained to ultimately reduce unsatisfied demand for contraception and achieve universal access to SRH services. With rising global poverty and the diversion of international family planning funding (as exemplified by the recent shortfall in the United Nations Population Funds' funding for family planning programmes), there is continued cause for concern over women's access to affordable contraceptive services in sub-Saharan Africa. Sustained global and local advocacy and monitoring are needed to ensure accountability and progress towards essential SRH service provision, including women's access to and use of contraception.

**Author affiliations**
[1]Department of Population, Family and Reproductive Health, Johns Hopkins University Bloomberg School of Public Health, Baltimore, Maryland, USA
[2]Soins et Santé Primaire, Centre for Research in Epidemiology and Population Health (CESP) U1018, INSERM, Paris, France
[3]University of Ibadan College of Medicine, Ibadan, Oyo, Nigeria
[4]School of Public Health, University of Kinshasa, Kinshasa, Republic of Congo
[5]Institut Supérieur des Sciences de la Population, Université de Ouagadougou, Ouagadougou, Burkina Faso
[6]International Centre for Reproductive Health, Mombasa, Kenya

**Contributors** CM is responsible for the overall content of this paper. FMO, PAk, GG, PG, LZ and PAn designed the PMA panel study. CM, FMO, PAk, GG, PG, LZ and PAn designed the COVID module. CM, CK and SW designed this particular study. CM conducted the analysis. CM and KW conducted the literature review. CM, KW, CK and SW wrote the initial draft. FMO, PAk, GG, PG, LZ and PAn interpreted the results and revised the manuscript.

**Funding** This research was developed under grants OPP1198333 and OPP1198339 awarded by the Bill & Melinda Gates Foundation to Johns Hopkins Bloomberg School of Public Health and Jhpiego.

**Competing interests** None declared.

**Patient and public involvement** Patients and/or the public were not involved in the design, or conduct, or reporting, or dissemination plans of this research.

**Patient consent for publication** Consent obtained directly from patient(s).

**Ethics approval** PMA surveys received approval from ethical committees in each geography (Burkina Faso: Comité d'Ethique Institutionnel Pour La Recherche en Santé, Ministère de l'Enseignement Supérieur, de la Recherche Scientifique et de l'Innovation; Democratic Republic of Congo: University of Kinshasa School of Public Health; Kenya: Kenyatta National Hospital–University of Nairobi Ethics Research Committee; Nigeria: Lagos State University Teaching Hospital Health Research Ethics Committee). PMA also received approval from the Institutional Review Board of the Johns Hopkins School of Public Health (IRB00014702). Public health preventive measures (ie, social distancing, mask wearing, use of hand sanitiser) were instituted in phase 2 to prevent COVID-19 transmission during data collection.

**Provenance and peer review** Not commissioned; externally peer reviewed.

**Data availability statement** Data are available in a public, open access repository. PMA data are publicly available at https://pmadata.org.

**ORCID iDs**
Caroline Moreau http://orcid.org/0000-0002-8637-6249
Shannon Wood http://orcid.org/0000-0003-4389-3526
Linnea Zimmerman http://orcid.org/0000-0002-0118-0889

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
