## [Reviewer comments · BMJ Open]

ARTICLE DETAILS

TITLE (PROVISIONAL)	Trends in fertility intentions and contraceptive practices in the context of COVID-19 in sub-Saharan Africa: Insights from four national and regional population-based cohorts
AUTHORS	Moreau, Caroline; Karp, Celia; Wood, Shannon; Williams, Kelsey; Olaolorun, Funmilola; Akilimali, Pierre; Guiella, Georges; Gichangi, Peter; Zimmerman, Linnea; Anglewicz, Philip

VERSION 1 – REVIEW

REVIEWER	Bolarinwa, Obasanjo University of KwaZulu-Natal, Department of Public Health Medicine
REVIEW RETURNED	27-Apr-2022

GENERAL COMMENTS	Trends in fertility intentions and contraceptive practices in the context of COVID-19 in sub-Saharan Africa: Insights from four national and regional population-based cohorts Abstract Line 9 – I am concerned about the statement, “We aimed to examine these trends in four African settings one year into the pandemic” this is speaking to an intention of doing something when the study has already been done. This same line is found between lines 32 and 33. Line 31 – Please change “COVID” to COVID-19 Conclusions The latter part of the line 36 conclusion is not in line with the results presented in the abstract. The study did not measure poverty, so there is no evidence to support rising poverty. The authors should endeavor to conclude and recommend based on the result of the study. Introduction The introduction is well written generally and provides a wonderful background to the study. However, I have a few suggestions that could make the section more interesting. To enhance this section, the authors should endeavor to provide information on existing policies or interventions tailored toward encouraging women to use contraceptives that were in place in these countries prior to the COVID-19 pandemic. The authors should provide brief information on COVID-19 and how the unprecedented occurrence generally affected SSA before tailoring it to SRH. I understand there is more information about this. However, this article is a whole and can be picked by someone in 1000 years; the potential future readers need to know what COVID-19 is in this article as well. Method The method is well detailed and covers the methodology required. However, I have a concern about how the sample size in Burkina Faso and Kenya could represent countries while DRC & Lagos were
---

	regions of countries – I noticed the sample sizes of DRC & Lagos is small, and the surveys were done in a location. This information on Burkina Faso and Kenya was not provided. Providing this information could inform if the authors could be silent with “Kinshasa” and “Lagos”. This concern could be a big one; why estimate for countries and regions simultaneously? Alignment of the study settings of the same characteristics is very important? I would suggest the authors minimize the use of “Kinshasa” and “Lagos” in all parts of the study and explain with justification in the method section only while the data was just limited to “Kinshasa” and “Lagos,” and this can also be included in the limitation section. I am aware that PMA data collection in the country is often not more than 4 regions, so there is no point specifying too much besides the study’s title being well detailed. Results This section is well detailed and self-explanatory. Discussion More African studies should be cited here, those conducted pre-COVID-19 and amid COVID-19. Limitation and strength The authors should endeavor to include this section in order to underscore all the limitations and uniqueness of the study Implications for policy and practice I recommend the authors to develop this section and use it to communicate to policymakers and NGOs on the implications of this study to practice and policy. Conclusion This section is missing
--	---

REVIEWER	Backhaus , Andreas Bundesinstitut für Bevölkerungsforschung
REVIEW RETURNED	01-Jun-2022

GENERAL COMMENTS	This paper uses recently released data from the Performance Monitoring for Action (PMA) to investigate whether fertility preferences, contraceptive behaviors, and unintended pregnancies in four sub-Saharan African countries have changed during the COVID-19 pandemic in comparison to a period shortly before the outbreak of the pandemic. These are timely questions with relevance for public health-related assessments of the impact of the pandemic on sub-Saharan African countries. Please find a number of remarks and suggestions below. CONTRIBUTION: As it sometimes happens when new data on an urgent topic have just been released, multiple researchers might be working on it. In this case, I may point out that a manuscript titled “Pregnancies and contraceptive use in four African countries during the COVID-19 pandemic”, of which I am the author, has already been accepted for publication in the Vienna Yearbook of Population Research. The accepted version can be found at SocArXiv (https://bit.ly/3NbUVku). However, I think the contributions of the two manuscripts are sufficiently different: My paper considers rates of pregnancy and modern contraceptive usage more broadly from a demographic perspective differentiated by age and education, but without differentiating between intended and unintended pregnancies, or contraceptive usage during the pandemic conditional on the pre-pandemic form of usage. Further, the underlying data and empirical approaches slightly differ, as I utilize also the pre-longitudinal PMA
---

	survey rounds for the pre-pandemic baseline and my results are estimated using linear probability models. Hence, while my results and the results presented in this manuscript are broadly in agreement with each other, I regard the two papers as rather complementary. However, the authors may consider delineating their contribution from my work in their manuscript. METHODS: While the various regression equations underlying the results do not necessarily have to be included in the main body of the manuscript, I suggest presenting and explaining them at least in the Appendix, as it would facilitate the interpretation of the results tables. RESULTS: In the Appendix tables 1 and 2, you appear to print all significant estimates in bold, which facilitates the tables' readability. I suggest following the same procedure in your results tables in the main body if permitted. The results regarding the rates of unintended pregnancy in Table 3 suggest a decline in unintended pregnancies in Burkina Faso by 0.7 percentage points or 25%, which is a large decline but not statistically significant. You may consider highlighting, for example, in the Discussion section, that the samples might be too small for reliably detecting even large changes in such relatively rare events. Related to my comment on the regression equations, in Table 4, it is not clear to me what the "Overall" estimate and the p values at the top of every panel refer to, and what the relationship of the p values to the individual estimates for the different classifications is. For example, when discussing Table 4, you write, "there were no differential trends by education, recent loss of income, or recent experience of food insecurity", but several estimates in the corresponding bottom panels of Table 4 appear to be significant based on their confidence intervals. Please add clarifications in this regard and also consider explanatory notes below the tables if permitted. DISCUSSION: The potential explanations for your results that you bring forward and discuss are thoughtful, competent, and consistent.
--	---

VERSION 1 – AUTHOR RESPONSE

Reviewer: 1

Dr. Obasanjo Bolarinwa, University of KwaZulu-Natal

Comments to the Author:

Trends in fertility intentions and contraceptive practices in the context of COVID-19 in sub-Saharan Africa: Insights from four national and regional population-based cohorts

Abstract

• Line 9 – I am concerned about the statement, "We aimed to examine these trends in four African settings one year into the pandemic" this is speaking to an intention of doing something when the study has already been done.

Response: We have changed the sentence as follows "This study examined trends in contraceptive behaviors in four sub-Saharan African settings one year into the pandemic." P2-Line5.

- Line 31 – Please change “COVID” to COVID-19 : We have made the change
- Conclusions The latter part of the line 36 conclusion is not in line with the results presented in the abstract. The study did not measure poverty, so there is no evidence to support rising poverty. The authors should endeavor to conclude and recommend based on the result of the study.
Response: The last sentence of the conclusion is a recommendation for future work based on the World Bank’s assessment of rising global poverty, not on our own assessment of poverty. We have revised the sentence as follows: “However, with reports of rising poverty in sub-Saharan Africa, there is continued need to monitor access to essential sexual and reproductive health services”.

The introduction is well-written generally and provides a wonderful background to the study. However, I have a few suggestions that could make the section more interesting. To enhance this section, the authors should endeavor to provide information on existing policies or interventions tailored toward encouraging women to use contraceptives that were in place in these countries prior to the COVID-19 pandemic.

The authors should provide brief information on COVID-19 and how the unprecedented occurrence generally affected SSA before tailoring it to SRH. I understand there is more information about this. However, this article is a whole and can be picked by someone in 1000 years; the potential future readers need to know what COVID-19 is in this article as well.

Response: Thank you for this feedback. We have revised the introduction as follows:

- We included an initial paragraph providing information about the general health and economic impact of COVID-19 during the study period (first year of the pandemic in 2020; page 4, lines 177-184).

“The COVID-19 pandemic, which started in late 2019, quickly traversed the globe, creating health and economic crisis. By the end of 2020, the SARS-CoV-2 virus had resulted in nearly two million deaths [1], destroyed 114 million jobs [2], and debilitated global economic growth rates, which fell to an annualized negative rate of -3.2 [2]. With 65,468 fatalities, the sub-Saharan continent bore less of a health burden in this early stage of the pandemic [1], but swift policy containment measures led to unprecedented economic loss, with a 1.9% contraction in the Gross Domestic Product (GDP) per capita and an estimated 32 million additional people living in extreme poverty [3]. “

- We also expanded on pre-pandemic sexual and reproductive health indicators in SSA to illustrate the preexisting family planning challenges. P4 lines 189-191

“In 2019, the unintended pregnancy rate (91 per 1000 women per year) in sub-Saharan Africa was the highest in the world [12] and only 55% of women’s need for contraception was satisfied [13].”

Method

The method is well detailed and covers the methodology required. However, I have a concern about how the sample size in Burkina Faso and Kenya could represent countries while DRC & Lagos were regions of countries – I noticed the sample sizes of DRC & Lagos is small, and the surveys were done in a location. This information on Burkina Faso and Kenya was not provided. Providing this information could inform if the authors could be silent with “Kinshasa” and “Lagos”. This concern could be a big one; why estimate for countries and regions simultaneously? Alignment of the study settings of the same characteristics is very important?

I would suggest the authors minimize the use of “Kinshasa” and “Lagos” in all parts of the study and explain with justification in the method section only while the data was just limited to “Kinshasa” and “Lagos,” and this can also be included in the limitation section. I am aware that PMA data collection in the country is often not more than 4 regions, so there is no point specifying too much besides the study’s title being well detailed.

Response: The PMA survey does not cover all provinces and states in DRC and Nigeria, but rather concentrates on specific geographies (Kinshasa province, Kongo Central Provinces in the DRC and Lagos State and Kano State in Nigeria). Therefore, unlike the nationally representative surveys conducted in other geographies, surveys implemented in the DRC and Nigeria are only regionally representative. We did not include Kongo Central and Lagos State in our analysis given smaller sample sizes and low prevalence of modern contraceptive use, resulting in very limited statistical power to show reductions in contraceptive use indicators over time.

We added the following sentence to clarify the regional versus national representative of PMA data across geographies, on page 5, lines 230-234: “Surveys are nationally representative in Burkina Faso

and Kenya, and regionally representative in the province of Kinshasa in the Democratic Republic of Congo and Lagos State in Nigeria. We did not include data from Kongo Central in the DRC and Kano State in Nigeria, also collected through the PMA platform due to smaller sample size and lack of statistical power to show any reduction in contraceptive use in these two geographies.”

We agree that the country of Burkina Faso and Lagos State are not directly comparable, given differences in rural/urban composition. Our objective is to show how women living in different geographies in SSA, which have different levels of economic development and healthcare systems, may have responded differently to the COVID-19 pandemic. In our previous paper, we stratified results by rural and urban areas in Burkina Faso and Kenya to compare the impact of COVID-19 across a diversity of urban and rural settings. We have added this sub-analysis as an appendix to offer additional insights on urban and rural trends across geographies. We found greater increases in contraception in rural than urban areas in Burkina Faso, but other trends were similar by urban/rural location in Burkina and Kenya.

We added the following sentence in the method section p 7 line 334-336 to describe the stratified analysis ‘We conducted the same analysis stratified by rural and urban residence in Burkina Faso and Kenya, to compare trends across urban localities (urban Burkina Faso, urban Kenya, Lagos_Nigeria and Kinshasa_DRC and rural localities in Burkina Faso and Kenya).’ Results of the stratified analysis are presented in Appendix 2.

Results

This section is well detailed and self-explanatory.

Response: Thank you for this comment. Based on previous question, we added results presentation to compare rural and urban trends across geographies (p 9 and 10)

Discussion

More African studies should be cited here, those conducted pre-COVID-19 and amid COVID-19.

Response: We have updated the discussion with recent literature from Africa about COVID-19's impact.

Limitation and strength

The authors should endeavor to include this section in order to underscore all the limitations and uniqueness of the study Implications for policy and practice I recommend the authors to develop this section and use it to communicate to policymakers and NGOs on the implications of this study to practice and policy.

Response: Thank you for this suggestion. We have extended the limitations section and have added a section on the strengths of the study. The added section is as follows:

P 15 lines 535-545 “Our data in DRC and Nigeria are not nationally representative but rather reflect urban trends. Stratified analyses comparing urban and rural Burkina Faso and Kenya found no major differences in family planning trends, however, these results cannot be generalized to DRC and Nigeria, given distinct patterns of SRH disruptions reported in several studies in Africa [25-26]. Finally, our sample sizes were limited to detect significant trends in rare events, such as pregnancy outcomes, or differential trends by women’s sociodemographic and economic background.”

P 16 lines 546-559 Our study also has several strengths. First, our population-based approach provides a complimentary perspective to data collected via Health Management Information Systems (HMIS), showing little change in behaviors and SRH outcomes, despite facility-based disruptions, suggesting the importance of non-facility-based services and self-care practices in ensuring continuity of family planning care [25]. Second, our novel study design, integrating perspectives from representative cross-sectional data and panel data, offers the double advantage of examining population and individual trends between pre- COVID-19 and COVID-19 periods, with the potential of identifying disparities in COVID-19 effects on family planning practices. The use of a comprehensive set of indicators ranging from pregnancy exposure to contraceptive practices and reproductive outcomes also provides a more holistic understanding of fertility processes during a pandemic, elucidating how family planning needs and use evolve concurrently. Finally, the cross-national/regional nature of the study, enhanced by using a common study protocol and indicators

across geographies strengthens comparative trends analysis, supporting previously reported variations in COVID-19 effects on SRH services [25-26].

Conclusion

This section is missing.

Response: Thank you for this comment. We revised the discussion section to include a conclusion. Specifically, we moved the last paragraph calling for sustained investment and monitoring of FP programs as the concluding paragraph of our paper

Reviewer: 2

Dr. Andreas Backhaus, Bundesinstitut für Bevölkerungsforschung

Comments to the Author:

CONTRIBUTION:

As it sometimes happens when new data on an urgent topic have just been released, multiple researchers might be working on it. In this case, I may point out that a manuscript titled "Pregnancies and contraceptive use in four African countries during the COVID-19 pandemic", of which I am the author, has already been accepted for publication in the Vienna Yearbook of Population Research.

The accepted version can be found at SocArXiv

(<https://nam02.safelinks.protection.outlook.com/?url=https%3A%2F%2Fbit.ly%2F3NbUVku&data=05%7C01%7Ccmoreau2%40jhu.edu%7C0db7fc2045494231c5d608da659f7051%7C9fa4f438b1e6473b803f86f8aedf0dec%7C0%7C0%7C637934033527593768%7CUnknown%7CTWFpbGZsb3d8eyJWIjoiMC4wLjAwMDAiLCJQIjoiV2luMzliLCJBTiI6IjEkaWwiLCJXVCi6Mn0%3D%7C3000%7C%7C%7C&data=23z4I3zr18AeNfKdm4aTC4znEqFV8slriW7x%2FUWat3M%3D&reserved=0>). However, I think the contributions of the two manuscripts are sufficiently different: My paper considers rates of pregnancy and modern contraceptive usage more broadly from a demographic perspective differentiated by age and education, but without differentiating between intended and unintended pregnancies, or contraceptive usage during the pandemic conditional on the pre-pandemic form of usage. Further, the underlying data and empirical approaches slightly differ, as I utilize also the pre-longitudinal PMA survey rounds for the pre-pandemic baseline and my results are estimated using linear probability models. Hence, while my results and the results presented in this manuscript are broadly in agreement with each other, I regard the two papers as rather complementary. However, the authors may consider delineating their contribution from my work in their manuscript.

Response: Thank you for sharing the manuscript. We have revised the introduction to acknowledge this previous analysis and highlight how the current paper adds to this work. P 4 lines 209-214

"Recent analysis estimating changes in modern contraception and pregnancy rates using repeated cross-sectional surveys collected face to face shed light on these trends. Specifically, in Burkina Faso, Kenya, Kinshasa (DRC), and Lagos (Nigeria), data collected in pre-pandemic periods and one year into the pandemic, suggest slight decreases in pregnancy rates among younger, less educated women in Burkina Faso and a rise in modern contraception among young women in all geographies, except Lagos [22]."

P 4 lines 216-222 "Expanding on these results, the present study describes population- and individual-level changes in women's fertility preferences, contraceptive behaviors, and unintended pregnancy experiences in the first year of the COVID-19 pandemic using cross-sectional representative data and panel data collected in the same sub-Saharan African geographies. Specifically, we examined how these trends differed by sociodemographic characteristics and COVID-19-related economic hardship. Finally, we assessed factors related to individual-level changes in contraceptive behaviors and to experiencing an unintended pregnancy during the pandemic.."

METHODS:

While the various regression equations underlying the results do not necessarily have to be included in the main body of the manuscript, I suggest presenting and explaining them at least in the Appendix, as it would facilitate the interpretation of the results tables.

Response: We have added an appendix with the GEE regression model (Appendix 1).

RESULTS:

In the Appendix tables 1 and 2, you appear to print all significant estimates in bold, which facilitates the tables' readability. I suggest following the same procedure in your results tables in the main body if permitted.

Response: We have indicated in bold, results for which confidence intervals do not overlap, our criteria for identifying significant trends.

The results regarding the rates of unintended pregnancy in Table 3 suggest a decline in unintended pregnancies in Burkina Faso by 0.7 percentage points or 25%, which is a large decline but not statistically significant. You may consider highlighting, for example, in the Discussion section, that the samples might be too small for reliably detecting even large changes in such relatively rare events. Response: Thank you for this suggestion. We added the following sentence to clarify this point. On page 15, lines 539-545: "Finally, our sample sizes were limited to detect significant trends in rare events, such as pregnancy outcomes, or differential trends by women's sociodemographic and economic background."

Related to my comment on the regression equations, in Table 4, it is not clear to me what the "Overall" estimate and the p values at the top of every panel refer to, and what the relationship of the p values to the individual estimates for the different classifications is. For example, when discussing Table 4, you write, "there were no differential trends by education, recent loss of income, or recent experience of food insecurity", but several estimates in the corresponding bottom panels of Table 4 appear to be significant based on their confidence intervals. Please add clarifications in this regard and also consider explanatory notes below the tables if permitted.

Response: The overall p-value refers to the significance of the interaction term between all categories of a demographic factor and the outcome of interest (e.g., contraceptive use, unintended pregnancy). However, we acknowledge some confidence intervals between specific categories do not overlap by age group in Burkina Faso, DRC, and Nigeria. Acknowledging the limited statistical power of interaction analysis (we have added this to the limitations in the discussion), we still discuss relevant patterns that are not statistically significant in the results section.

VERSION 2 – REVIEW

REVIEWER	Backhaus , Andreas Bundesinstitut für Bevölkerungsforschung
REVIEW RETURNED	14-Sep-2022

GENERAL COMMENTS	Thank you for addressing my comments and suggestions from the previous reviewer report. I may follow up with a short list of minor comments and suggestions that should contribute to the finalization of your manuscript:  - I do not intend to criticize you for something that a different reviewer asked you to do – but maybe you could add "confirmed" to the number of global deaths and fatalities in sub-Saharan Africa due to COVID-19 in the newly added first paragraph of the Introduction? It is generally accepted that deaths from COVID-19 were undercounted during the early stages of the pandemic and that there is high uncertainty about the COVID-19 death count in African countries. - Thank you for explaining the meaning of the p-values reported in Table 4. You might consider adding a note below the table with a similar explanatory text. - Fertility intentions seem to be missing from the added "Strengths and Limitations" bullet points – is this intentional? If not, you could mention them here. - I may suggest updating reference 22 to the following: Backhaus, A. Pregnancies and contraceptive use in four African
--

	countries during the COVID-19 pandemic. Vienna Yearbook of Population Research 2022;20. doi:10.1553/populationyearbook2022.dat.4  - Line 354: Might a formulation like “No significantly different trends were observed between women in rural and urban settings” express more clearly what you intend to say? - Line 468: You might consider mentioning that your findings on contraceptive usage are in accordance with the findings in Backhaus (2022). - Line 477: There appears to be an unresolved comment from one of the authors in the document. - References, line 37: The numbering of Wood et al. 2021 seems to be out of step with the rest of the References.
--	---

VERSION 2 – AUTHOR RESPONSE

Reviewer:

- I do not intend to criticize you for something that a different reviewer asked you to do – but maybe you could add “confirmed” to the number of global deaths and fatalities in sub-Saharan Africa due to COVID-19 in the newly added first paragraph of the Introduction? It is generally accepted that deaths from COVID-19 were undercounted during the early stages of the pandemic and that there is high uncertainty about the COVID-19 death count in African countries.

We agree with this reviewer’s comment and have added the word “confirmed” in the following sentence “With 65,468 confirmed fatalities, the sub-Saharan continent bore less of a health burden in this early stage of the pandemic [1], but swift policy containment measures led to unprecedented economic loss, with a 1.9% contraction in the Gross Domestic Product (GDP) per capita and an estimated 32 million additional people living in extreme poverty [3].”

- Thank you for explaining the meaning of the p-values reported in Table 4. You might consider adding a note below the table with a similar explanatory text.

We have added the following note below Table 4 : “*p-values refer to the significance of the interaction term between all categories of a demographic factor and the outcome of interest (contraceptive use)”

- Fertility intentions seem to be missing from the added “Strengths and Limitations” bullet points – is this intentional? If not, you could mention them here.

Thank you for suggestion. We have added the following sentences to discuss fertility intentions in the strength and limitation sections.

Limitations: “In addition, a more thorough investigation of changing fertility intentions and their impact on subsequent reproductive outcomes over the course of the pandemic could shed more light on the ways fertility responds to external shocks.”

Strength: In particular, our focus on unintended pregnancy provides added information about women’s ability to achieve their reproductive goals in the context of the global pandemic.

- I may suggest updating reference 22 to the following:

Backhaus, A. Pregnancies and contraceptive use in four African countries during the COVID-19 pandemic. Vienna Yearbook of Population Research 2022;20.

doi:10.1553/populationyearbook2022.dat.4

Thank you for this information, we have updated the reference

- Line 354: Might a formulation like “No significantly different trends were observed between women in

rural and urban settings” express more clearly what you intend to say?

We have revised the sentence as follows: “No significant differences in trends were observed between women in urban and rural settings (Appendix 2).”

- Line 468: You might consider mentioning that your findings on contraceptive usage are in accordance with the findings in Backhaus (2022).

We agree and have added the following sentence in the 2nd paragraph of the discussion “The absence of negative population-level effects of the pandemic on contraceptive behaviors and reproductive outcomes, noted in a previous study focusing on the early months of the pandemic [14,15], and confirmed in a study using one year follow data [22] are surprisingly reassuring and suggest several potential explanations [23].”

- Line 477: There appears to be an unresolved comment from one of the authors in the document.

We are sorry for not spotting this mistake. The clean version had updated the missing reference.

“First, our population-based approach provides a complimentary perspective to data collected via Health Management Information Systems (HMIS), showing little change in behaviors and SRH outcomes, despite facility-based disruptions, suggesting the importance of non-facility-based services and self-care practices in ensuring continuity of family planning care [25].”

- References, line 37: The numbering of Wood et al. 2021 seems to be out of step with the rest of the References.

We have fixed this mistake. The Wood reference is now correctly mentioned as reference 14